# Impaired bone quality characterized by apatite orientation under stress shielding following fixing of a fracture of the radius with a 3D printed Ti-6Al-4V custom-made bone plate in dogs

Keiichiro Mie[1], Takuya Ishimoto[2], Mari Okamoto[1], Yasumasa Iimori[1], Kazuna Ashida[1], Karin Yoshizaki[1], Hidetaka Nishida[1], Takayoshi Nakano[2], Hideo Akiyoshi[1]*

1 Laboratory of Veterinary Surgery, Division of Veterinary Science, Course of Veterinary Science, Graduate School of Life and Environmental Sciences, Osaka Prefecture University, Izumisano, Osaka, Japan,
2 Division of Materials and Manufacturing Science, Graduate School of Engineering, Osaka University, Suita, Osaka, Japan

* akiyoshi@vet.osakafu-u.ac.jp

**Data Availability Statement:** All relevant data are within the manuscript.

## Abstract

Custom-made implants have recently gained attention in veterinary medicine because of their ability to properly fit animal bones having a wide variety of shapes and sizes. The effect of custom-made implants on bone soundness and the regeneration process is not yet clear. We fabricated a 3D printed Ti-6Al-4V custom-made bone plate that fits the shape of the dog radius, and placed it into the radius where an osteotomy had been made. The preferential orientation of the apatite $c$-axis contributes to the mechanical integrity of the bone and is a reliable measure of bone quality. We determined this parameter as well as the bone shape and bone mineral density (BMD). The bone portion which lies parallel to the bone plate exhibited bone resorption, decreased BMD, and significant degradation of apatite orientation, relative to the portion outside the plate, at 7 months after the operation. This demonstrates the presence of stress shielding in which applied stress is not transmitted to bone due to the insertion of a stiff bone plate. This reduced stress condition clearly influences the bone regeneration process. The apatite orientation in the regenerated site remained different even after 7 months of regeneration, indicating insufficient mechanical function in the regenerated portion. This is the first study in which the apatite orientation and BMD of the radius were evaluated under conditions of stress shielding in dogs. Our results suggest that assessment of bone repair by radiography can indicate the degree of restoration of BMD, but not the apatite orientation.

## Introduction

Bone fractures of the radius and ulna are one of the most frequent injuries in dogs [1]. In particular, toy breed dogs often fracture the radius and ulna after jumping or falling, and the

**Funding:** This work was supported by the Cross-Ministerial Strategic Innovation Promotion Program (SIP) titled "Materials Integration for Revolutionary Design System of Structural Materials" of the Japan Science and Technology Agency (JST), and "Innovative Design/Manufacturing Technologies" of the New Energy and Industrial Technology Development Organization (NEDO).

**Competing interests:** The authors have declared that no competing interests exist.

fracture frequently leads to complications such as delayed union, nonunion, lameness, or refracture [1–3]. Treatment options for bone fractures of the radius include various surgical techniques; open reduction with bone plate fixation is the most common treatment in dogs with radius fractures [1, 4].

Custom-made implants are becoming increasingly important in veterinary medicine for better bone reconstruction [5, 6]. Variations in skeletal size and shape between animals and breed are much larger than that among human patients. It may therefore be more desirable to apply an implant that matches the individual skeleton than to select a similar size from standard products, so as to prevent mismatches in size and shape between the patient skeleton and the implant. The bone regeneration process and bone soundness following a custom-made implant to a fractured site has not been sufficiently evaluated in veterinary medicine.

Regeneration of bone fracture sites has generally been evaluated by radiography. Radiography can determine the degree of morphological regeneration at a bone fracture site, but cannot evaluate the extent of regeneration of bone mechanical function. Bone mechanical function can be attributed to the orientation of the mineral and organic constituent phases of biological apatite crystals and type I collagen fibers [7–11]. Bone apatite and collagen have specific crystallographic textures which depend on the type of bone [12], and apatite material exhibits anisotropy in its intrinsic mechanical properties, such as Young's modulus [13, 14]. Recently, the degree of directionality of the apatite $c$-axis has been recognized as an indicator of bone mechanical function [7, 9, 11], because the Young's modulus along the $c$-axis is greater than along the $a$-axis [13, 14]. Indeed, the preferential orientation of apatite is reported to decrease in some abnormal conditions, degrading the mechanical function of the bone [15–18]. Apatite orientation is therefore a potentially important index of bone quality in veterinary medicine.

The most important factor affecting apatite orientation and related mechanical function is mechanical stress experienced by the bone. The apatite orientation of regenerated bone depends on the magnitude of the applied stress, for instance [7]. Moreover, the orientation deteriorates where the stress is reduced [19]. Bone plate fixation is a typical bone fracture treatment in veterinary medicine, and reduces mechanical stress, a phenomenon known as stress shielding [20, 21]. Stress shielding induced by insertion of the bone plate can cause refracture of the bone after the plate is removed in humans [22], which might be at least partly responsible for degradation of the apatite orientation along the principally direction of loading [23]. In dogs, stress shielding has been recognized as a cause of complications such as delayed union or refracture. The preferential apatite orientation has been studied in dogs with hip implants [24] or tooth root implants [25], but no studies have been made of restoration of the preferential apatite orientation at the site of a bone fracture treated with a bone plate in dogs.

In the present study, the preferential orientation of the apatite $c$-axis was determined in dogs with an experimental bone fracture of the radius, which was fixed with a custom-made 3D printed bone plate. The restoration of the bone mineral density (BMD) was also studied and compared with the apatite orientation.

## Materials and methods

### Experimental animals

Four healthy beagles (females, 2–5 years old, 7.8–10.8 kg) were used in this study. The dogs belonged the kennel of Research Center for Experimental Animal Science of Osaka Prefecture University. The dogs were considered healthy based on their medical history and physical examination. The study was conducted according to the guidelines of the Experimental Animal Committee of Osaka Prefecture University and was approved by the Experimental Animal Committee of Osaka Prefecture University (Permit number 29–104).

## Planning and printing of implant

Custom-made 3-dimensional (3D) printed bone plates were designed using *in silico* analysis of images from preoperative computed tomography (CT) of a beagle. The data was acquired using a 16-slice scanner at 0.5-mm intervals (Activion 16, Toshiba, Tokyo, Japan). These high-resolution images were transmitted to the medical device manufacturer (Athena Pet, Soft Cube, Osaka, Japan), where the virtual cages and lattice structures were transmitted into a 3D computer-aided design (CAD) model. After the patient's CT data had been integrated with the CAD model, the shape of the bone plate was modified to match the distal part of the radial diaphysis of the dog, and implant placement was simulated *in silico* to check the accuracy. Next, 3D stereolithography (STL) print files were prepared from the CAD models (Fig 1A), from which the implants were built through a powder bed type 3D printer. Medical-grade Ti-6Al-4V ELI (Extra Low Interstitials) powder was purchased from EOS (Munich, Germany), and the bone plate was fabricated in a layer-by-layer manner on the basis of the STL model using a selective laser melting (SLM) system (M 290, EOS, Munich, Germany). The custom-made 3D printed bone plate is shown in Fig 1B.

## Experimental procedure

The dogs were injected subcutaneously with atropine sulfate (0.025 mg/kg, Fuso Pharmaceutical Industries, Osaka, Japan) and robenacoxib (0.2 mg/ kg, Onsior, Elanco Japan, Tokyo, Japan) before anesthesia. General anesthesia was induced by intravenous administration of propofol (6 mg/kg, Intervet, Osaka, Japan), following intravenous injection of midazolam (0.2 mg/kg, Dormicum, Astellas Pharma, Tokyo, Japan) and butorphanol (0.2 mg/kg, Vetorphale, Meiji Seika Pharma, Tokyo, Japan). An endotracheal tube was placed into the trachea to facilitate control of respiration. Anesthesia was maintained with 1.5–2.0% isoflurane (Mylan Seiyaku, Tokyo, Japan) and oxygen. Cefazolin (30 mg/kg, Cefamezin α, LTL Pharma, Tokyo, Japan) was administered intravenously prior to the operation. An incision was made through the skin and subcutaneous tissue so as to expose the radial diaphysis. An experimental transverse fracture of the right or left radial diaphysis was made with a surgical micro saw (Nakanishi, Tochigi, Japan) at the distal position of the radial diaphysis, and then was fixed using the

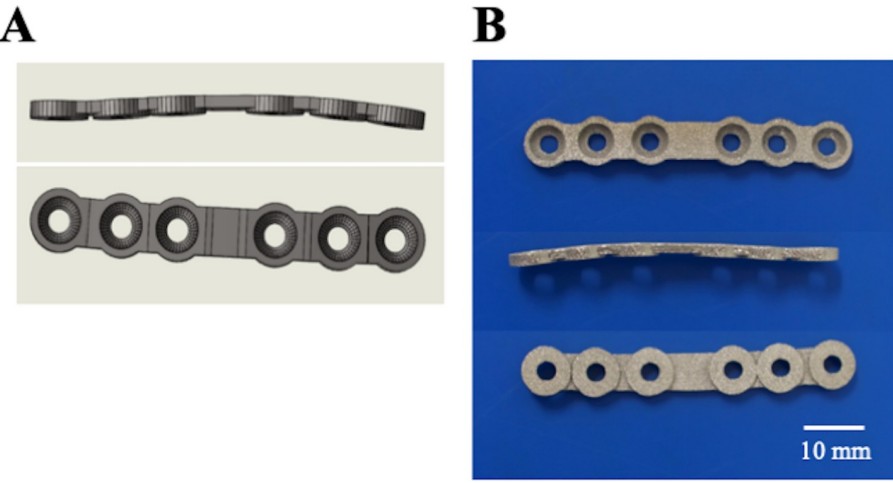

**Fig 1. Custom-made 3D printed bone plate.** (A) The shape of the designed bone plate. (B) The custom-made 3D printed bone plate. The plate was designed to contact the bone only via the parts of the screw holes. The screw holes were designed as a hemispherical shape fit for the screw head.

custom-made 3D printed bone plate and titanium screws (Platon Japan, Tokyo, Japan). Subcutaneous tissue and skin closure were carried out by routine procedures using 4–0 synthetic absorbable suture materials (Monosyn, B. Braun Aesculap Japan, Tokyo, Japan) and skin staples (Manipler AZ, Alfresa Pharma, Osaka, Japan). After the operation, the dogs were treated with cefazolin and meloxicam (0.1 mg/kg, Boehringer Ingelheim Animal Health Japan, Tokyo, Japan). No casts were applied after the operation.

The dogs were allowed to freely perform weight-bearing activities in the cage after the operation. Radiographs of the forelimb were taken at 1, 3, 5 and 7 months after the operation for assessment of the healing process of the fracture of the radius (Fig 2). The dogs were sacrificed by an intravenously administered overdose of potassium chloride with deep anesthesia at 7 months after the operation.

## Radiography of the removed radii

The radii were removed with the neighboring ulnas and immersed in 70% ethanol. Soft X-ray photographs were taken using XIE (Chubu Medical, Mie, Japan), at 30 kV and 30 μA radiation (Fig 3).

## Bone morphology observation

The screws and bone plate were removed with care and the radius was isolated from the neighboring ulna. Micro-computed tomography (μCT) (SMX-100CT, Shimadzu, Kyoto, Japan) was used to produce bone images with a spatial resolution of 57 μm on each side, so as to observe the cross-sectional morphology of the radius. Cross-sectional images were produced at 10 sites (positions 1 to 10; see Fig 4A).

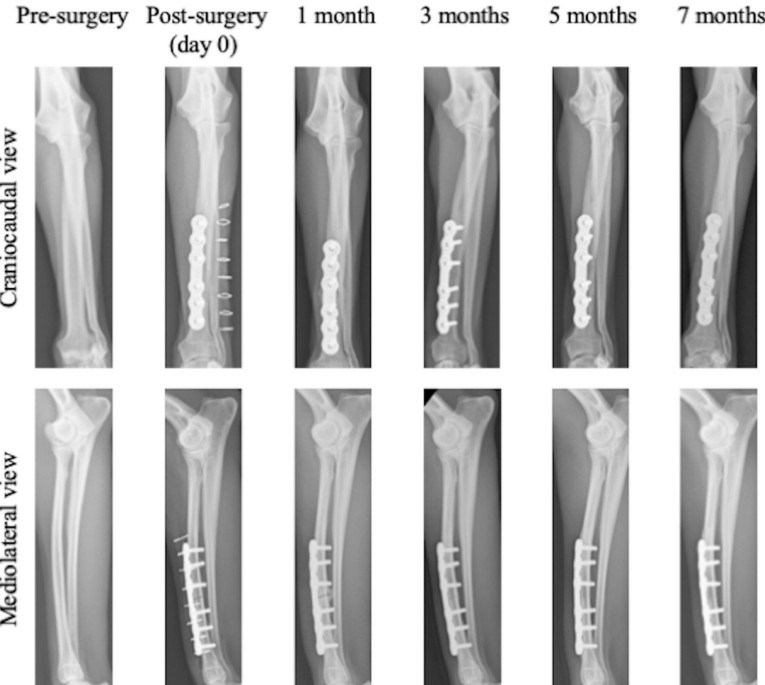

**Fig 2. Representative craniocaudal and mediolateral radiographic views before and after surgery.** Radiographs of forelimb taken pre-surgery, post-surgery (day 0), and 1, 3, 5 and 7 months after the operation.

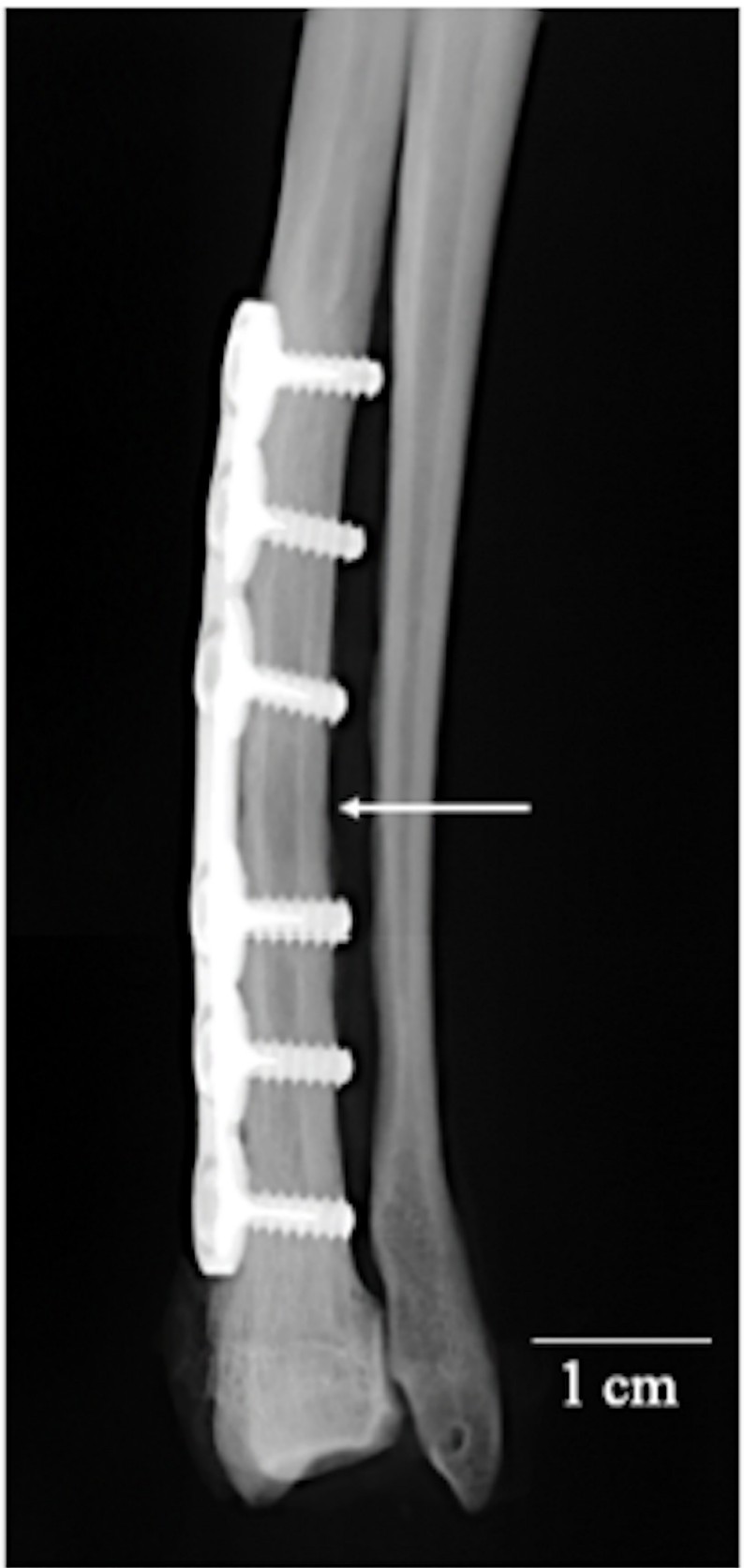

**Fig 3. Radiographs of forelimb with a bone plate at 7 months after the operation.** An arrow indicates the osteotomized position.

## Evaluation of BMD

Volumetric BMD was measured at 10 points (positions 1 to 10; see Fig 4A) in the radius in 70% ethanol, using an XCT Research SA+ system (Stratec Medizintechnik, Birkenfeld, Germany) at 50.7 kV and 0.276 mA with a resolution of $80 \times 80 \times 460$ μm. The BMD of the cortical bone region having a BMD value of 690 mg/cm$^3$ or higher [26] were calculated.

## Analysis of apatite c-axis orientation

The apatite *c*-axis orientation was analyzed by a microbeam X-ray diffractometer (μXRD) system (R-Axis BQ, Rigaku, Tokyo, Japan) equipped with a transmission-type optical system, and an imaging plate (storage phosphors) (Fuji Film, Tokyo, Japan) placed behind the specimen. Mo-Kα radiation of wavelength 0.07107 nm was generated at a tube voltage of 50 kV and tube current 90 mA. The distance between the detector and the X-ray focus of the specimen was 127.4 mm. The pixel size of the imaging plate was 100 μm × 100 μm. The incident beam was focused on a beam spot of diameter 800 μm by a double-pinhole metal collimator and radiated vertically to the long axis of the bone so as to capture diffraction information along the bone axis. The incident X-ray was transmitted along the craniocaudal axis from the cranial surface; diffraction data were collected for 180 s.

From the resulting diffraction intensity pattern (Debye ring) (see Fig 5), the two representative diffraction peaks for apatite, (002) and (310), were used to analyze the apatite *c*-axis orientation, as described previously [15, 19]. In long bones, the apatite *c*-axis orients preferentially along the bone longitudinal axis [12] aligned with the collagen matrix [27]. We therefore analyzed the diffraction information along the long axis of the radius. The upper and lower parts of the Debye ring correspond to the radial long axis. Diffraction intensities were integrated azimuthally over a range of 100 pixels to obtain an X-ray diffraction profile. The degree of preferential orientation of the *c*-axis in the apatite crystals was determined as the relative intensity ratio of the (002) diffraction peak to the (310) peak in the X-ray profile. This is an appropriate

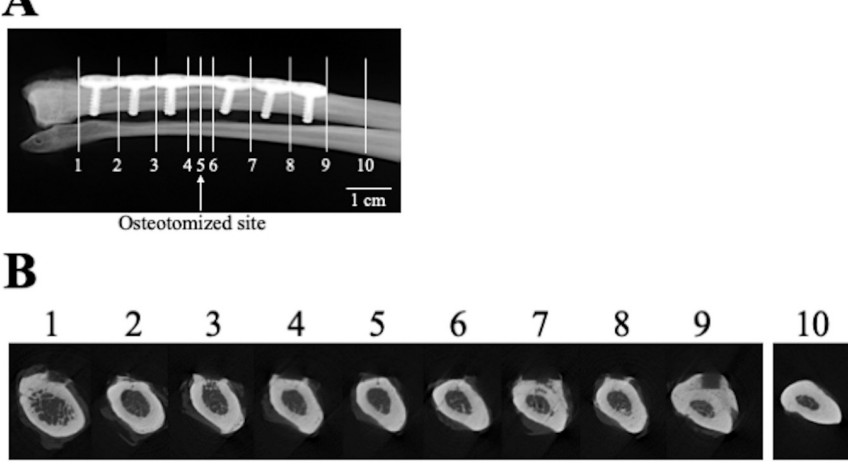

**Fig 4. μCT examination of the removed radii.** (A) Positions at which μCT images were taken. (B) μCT cross-sections at each position, 7 months after the operation.

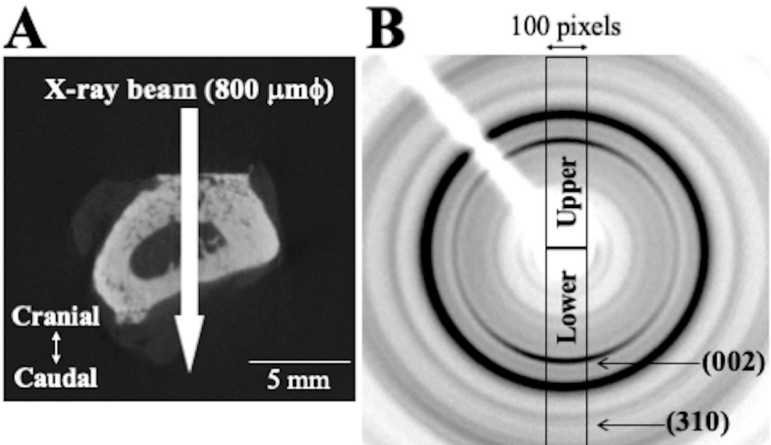

**Fig 5. Method for analyzing apatite *c*-axis orientation.** (A) Beam path of X-ray. (B) μXRD patterns (Debye rings); the vertical direction (upper and lower parts) corresponds to the radial long axis.

index for evaluating apatite orientation [12, 28] and bone mechanical function (Young's modulus) [7]. The intensity ratios calculated from the upper and lower parts of the Debye ring were averaged. Randomly-oriented hydroxyapatite (NIST #2910: calcium hydroxyapatite) powder had an intensity ratio of 0.8; consequently, detected values >0.8 indicate definite anisotropic apatite *c*-axis orientation in the radial long axis.

## Statistical analysis

All data are presented as mean ± standard deviation (SD). Statistical comparisons were performed using one-way ANOVA and Tukey's multiple comparison test. A value of $P < 0.05$ was considered statistically significant. All statistical analyses used SPSS 25 (SPSS Japan Inc., Tokyo, Japan) for Microsoft Windows.

## Results

In all dogs, fractures of the radial diaphysis were fixed with the custom-made 3D printed bone plates as planned. None of the dogs exhibited any abnormalities in gait at 1 week after the operation. Fig 2 shows radiographs of the forelimb. Callus formation was observed at 1 month after the operation, and no fracture line could be observed at 5 months after the operation in any dog. Fig 3 shows a high-resolution radiograph taken at 7 months postoperatively. The fracture line is completely invisible, and reconstruction of the cortical bone and medullary marrow cavity is evident. μCT cross-sectional images (Fig 4) also reveal formation of cortical bone and marrow cavity at the osteotomized site (position 5 in Fig 4B).

The BMD and the degree of preferential apatite *c*-axis orientation of the radii at 7 months after the operation are shown in Fig 6. In bone positions 1 to 9 which lie parallel to bone plate, either BMD or the degree of preferential apatite orientation was significantly less than in bone position 10, which is located outside the plate. The BMD values of the regenerated site (position 5) were comparable to those of the surrounding bone site (Fig 6A). The preferential apatite *c*-axis orientation in the regenerated site (position 5) was significantly degraded (Fig 6B), indicating that recovery of apatite orientation is not synchronized with that of BMD. Fig 7 schematically illustrates the variations in the apatite orientation and BMD in the intact portion without influences of stress shielding, in the stress shielded portion beneath the bone plate, and in the regenerated portion under stress shielding conditions.

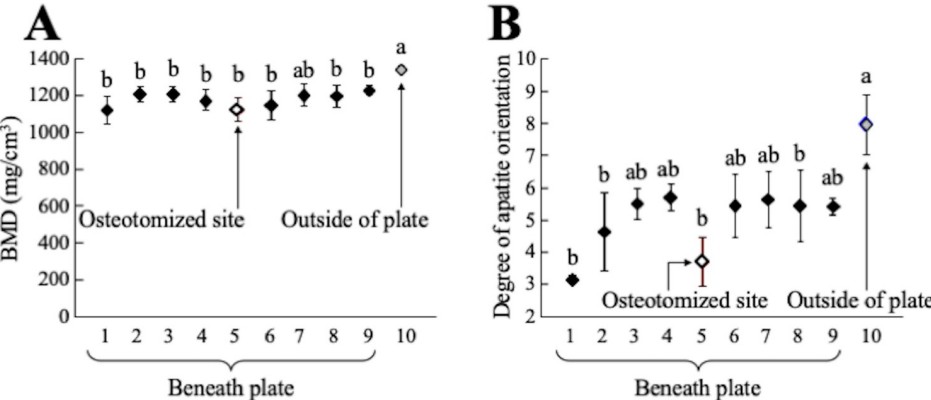

**Fig 6. BMD and preferential apatite *c*-axis orientation of the radii analyzed at the position shown in Fig 4A.** (A) Variation in BMD and (B) variation in the degree of preferential apatite *c*-axis orientation along the radial long axis as a function of bone position. Gray and white symbols represent the value in the intact (free from stress shielding) and regenerated positions, respectively. a: *P* < 0.05 vs position 5. b: *P* < 0.05 vs position 10.

## Discussion

In this study, we evaluated the changes and restoration of bone density and bone quality in the canine radius with a bone fracture fixed with a custom-made Ti-6Al-4V 3D printed bone plate. The custom-made plate appeared to cause stress shielding on the bone portion lying parallel to the stiff bone plate. A decrease in BMD and deterioration of bone quality occurred beneath the bone plate, and impaired restoration of bone quality was observed in the osteotomized site.

In the non-osteotomized site beneath the custom-made bone plate, significant degradation of bone material, characterized by decreased BMD and apatite orientation, was found. Moreover, bone resorption was observed in some μCT cross-sectional images (positions 3 and 7 in Fig 4B). These bone degenerative changes clearly indicate the presence of stress shielding. Previous studies have determined that reduced axial stress degrades the preferential apatite *c*-axis orientation in the loading direction of long bones [19, 23]. The apatite orientation, which is a recognized bone quality parameter, correlates with bone strength more strongly than BMD in the regenerated and pathological bones [7, 11]. Our results therefore indicate that bone

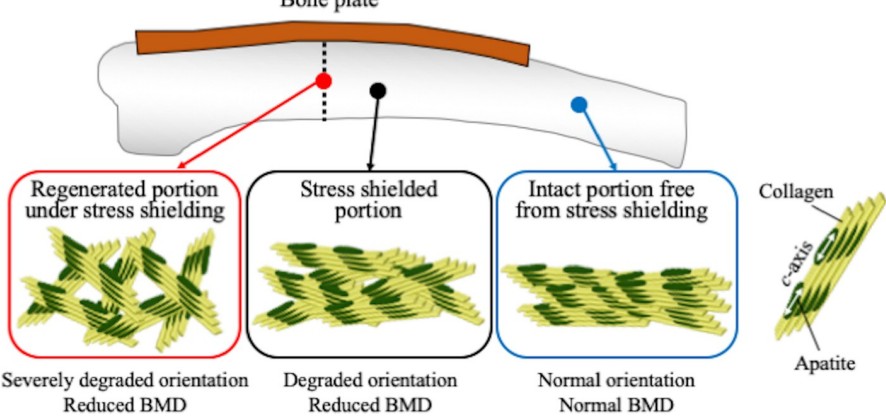

**Fig 7. Schematic drawing showing the variations in the apatite orientation and BMD in the intact, in the stress shielded, and in the regenerated portions under stress shielding conditions.**

strength deteriorates more severely than is expected from the decrease in BMD by stress shielding.

In the osteotomized site (position 5 in Fig 6), the BMD recovered to the same extent that the surrounding portion suffered from stress shielding. In contrast, the apatite orientation along the bone axis had scarcely recovered even after 7 months. It has been reported that the restoration of apatite orientation takes longer than the recovery of BMD, but, in the rabbit fracture model, the apatite orientation and bone strength was normalized after 6 months [7]. In the early stage of bone regeneration, a bone (callus) is formed that is less oriented [7, 28] and less mechanically integrated [7]. The formation of the bone marrow cavity and cortical bone in the regenerated site proves that bone replacement has taken place by remodeling activity, which is necessary to restore the apatite orientation to the normal condition [7]. The unrecovered apatite orientation observed in the regenerated portion would be caused by bone regeneration under the reduced stress conditions due to bone plate implantation. In other words, stress shielding as a result of a custom-made 3D printed bone plate is liable to prevent the restoration of preferential apatite orientation at a bone fracture site, and might cause refracture [22]. This unrecovered apatite orientation under stress shielding might be influenced by the osteon orientation. The orientation of osteon in which collagen preferentially orients along its longitudinal axis [29] corresponds with the directions of the maximum principal stress [30, 31]. Under the stress shielding, the direction of osteon would be disturbed, which further led to the low degree of collagen orientation. Since the apatite crystallizes on collagen so that its *c*-axis aligns with the long axis of collagen in the presence of osteocalcin [17], the apatite *c*-axis orientation degrades by inheriting degraded collagen orientation. With physiological stress during bone regeneration, the apatite orientation is reliably restored [7, 28]. The present study determined, for the first time, the apatite orientation and BMD of the radius under stress shielding conditions in dogs. Our results suggest that radiography can evaluate BMD but not the extent of restoration of the apatite orientation.

We used a custom-made 3D printed bone plate to fix the bone fracture of the radius; the bone plates were made from medical-grade Ti-6Al-4V powder, which is a general material available for bone implants. Our results indicate that Ti-6Al-4V implants may not be suitable for restoration of apatite orientation at bone fracture sites, although radiographic findings suggest restoration of bone fracture in dogs. For restoration of apatite orientation at a bone fracture site, it is possible that commercial or custom-made implants made from general materials, including pure titanium and Ti-6Al-4V alloy, are too stiff to exert adequate anisotropic stress on a fracture site in dog bones. Bone implants with less rigidity may be better for suppressing stress shielding [32], leading to better restoration of apatite orientation and mechanical function at a bone fracture site. Future studies should involve low-rigidity materials so as to determine their benefit in reducing bone degradation due to stress shielding. The most promising strategy for both suppression of stress shielding and shape customization is the creation of beta-type Ti alloy-single crystalline implants, utilizing a SLM 3D printer [33]. The Young's modulus of a single crystalline beta-type Ti-15Mo-5Zr-3Al alloy has an anisotropic Young's modulus; the lowest Young's modulus is very close to that of bone along the specific crystallographic direction [34]. In addition, SLM can provide implants adapted to the bone shape of each patient [35], as in the present.

This study has some limitations. First, apatite orientation in the regenerated radii was examined only at 7 months after fracture fixation. To estimate the appropriate rigidity of bone implants, further study would helpful to determine the change with time of the apatite orientation at a bone fracture site. Second, because anisotropic stress on the forearm is divided between the radius and the ulna, the influence of the ulna on support of skeletal anisotropic stress was not taken into account in this study.

In conclusion, fracture fixation with a custom-made 3D printed bone plate failed to restore apatite orientation at the osteotomized site of the radius in a dog, although restoration of BMD at the bone fracture site recovered to match the surrounding portion.

## Author Contributions

**Formal analysis:** Takuya Ishimoto, Takayoshi Nakano.

**Funding acquisition:** Takayoshi Nakano.

**Investigation:** Keiichiro Mie, Mari Okamoto, Yasumasa Iimori, Kazuna Ashida, Karin Yoshizaki, Hideo Akiyoshi.

**Methodology:** Takuya Ishimoto, Takayoshi Nakano.

**Supervision:** Takayoshi Nakano, Hideo Akiyoshi.

**Writing – original draft:** Keiichiro Mie, Takuya Ishimoto.

**Writing – review & editing:** Takuya Ishimoto, Hidetaka Nishida, Takayoshi Nakano, Hideo Akiyoshi.

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
