## [Decision Letter · Decision Letter 0]

24 Jul 2020

PONE-D-20-18343

Impaired bone quality characterized by apatite orientation under stress shielding following fixing of a fracture of the radius with a 3D printed Ti-6Al-4V custom-made bone plate in dogs

PLOS ONE

Dear Dr. Akiyoshi,

Thank you for submitting your manuscript to PLOS ONE. After careful consideration, we feel your manuscript is ready for publication after a minor revision.  Please submit a revised version of the manuscript that addresses the points raised by Reviewer 1.

We look forward to receiving your revised manuscript.

Kind regards,

Ryan K. Roeder, PhD

Academic Editor

PLOS ONE

Journal Requirements:

2. In your Methods section, please provide additional details regarding the dogs used in your study and ensure you have described the source. For more information regarding PLOS' policy on materials sharing and reporting, see https://journals.plos.org/plosone/s/materials-and-software-sharing#loc-sharing-materials.

3. Thank you for including your ethics statement:  "The study was conducted according to the guidelines of the Experimental Animal Committee of Osaka Prefecture University (Permit number 29–104).".   

Please amend your current ethics statement to confirm that your named ethics committee specifically approved this study.

For additional information about PLOS ONE submissions requirements for ethics oversight of animal work, please refer to http://journals.plos.org/plosone/s/submission-guidelines#loc-animal-research  

Reviewers' comments:

Reviewer's Responses to Questions

**Comments to the Author**

1. Is the manuscript technically sound, and do the data support the conclusions?

Reviewer #1: Yes

Reviewer #2: Yes

2. Has the statistical analysis been performed appropriately and rigorously? 

Reviewer #1: Yes

Reviewer #2: Yes

3. Have the authors made all data underlying the findings in their manuscript fully available?

Reviewer #1: Yes

Reviewer #2: Yes

4. Is the manuscript presented in an intelligible fashion and written in standard English?

Reviewer #1: Yes

Reviewer #2: Yes

5. Review Comments to the Author

Reviewer #1: Abstract: "flexibly" should be "properly"

Abstract: "remained impaired" should be "remained different" You have no evidence of impairment, just differences in axis orientation.

Line 149: "cares" should be "care"

General comment: The remodeling of fracture callus (woven bone) into lamellar bone is affected by loading since osteonal orientation is affected by bone stress (see, for example, "Skeletal Tissue Mechanics Edition 2" Martin et al, discussion around Figure 6.14). A discussion of this is not necessary, but would make the results of the paper fit better into existing literature. My conclusion from your experiment is that stress shielding under a plate that is insufficient to cause an effect on BMD, can still affect the details of consolidation of the woven bone of the fracture callus (i.e., collagen and mineral long-axis directions) by affecting paths of osteonal cutting cones and, perhaps, how the osteoblasts deposit matrix.

Reviewer #2: The authors presented a high-quality study of the effect of a single fracture plate for use in veterinary applications. The authors demonstrated reduced BMD and, more so, reduced apatite c-axis preferential orientation in the vicinity of the fracture site and corresponding surrounding bone plate. This reviewer does not have any additional questions.

6. PLOS authors have the option to publish the peer review history of their article (what does this mean?). If published, this will include your full peer review and any attached files.

Reviewer #1: No

Reviewer #2: No

---

## [Author Response · Author response to Decision Letter 0]

28 Jul 2020

Thank you for your valuable comments regarding our manuscript. In response to the reviewers’ comments, we have revised our manuscript as described below. In the revised manuscript and this response letter, the revised sections are marked in red color.

Editor:

Comments 1-3

Answer to comments 1

Thank you for kindly comment. We have revised the manuscript and figure files naming according to PLOS ONE's style templates.

2. In your Methods section, please provide additional details regarding the dogs used in your study and ensure you have described the source. For more information regarding PLOS' policy on materials sharing and reporting, see https://journals.plos.org/plosone/s/materials-and-software-sharing#loc-sharing-materials.

Answer to comments 2

Thank you for kindly comment. According to Editor’s comment, we have added the following sentences in Materials and Methods section.

P. 4, line 83

The dogs belonged the kennel of Research Center for Experimental Animal Science of Osaka Prefecture University. The dogs were considered healthy based on their medical history and physical examination.

3. Thank you for including your ethics statement: "The study was conducted according to the guidelines of the Experimental Animal Committee of Osaka Prefecture University (Permit number 29–104).". 

Please amend your current ethics statement to confirm that your named ethics committee specifically approved this study.

For additional information about PLOS ONE submissions requirements for ethics oversight of animal work, please refer to http://journals.plos.org/plosone/s/submission-guidelines#loc-animal-research

Once you have amended this/these statement(s) in the Methods section of the manuscript, please add the same text to the “Ethics Statement” field of the submission form (via “Edit Submission”)."

Answer to comments 3

Thank you for kindly comment. According to Editor’s comment, we have revised the following sentences about ethics statement in Materials and Methods section.

P. 4, line 85

The study was conducted according to the guidelines of the Experimental Animal Committee of Osaka Prefecture University and the protocol was approved by the Experimental Animal Committee of Osaka Prefecture University (Permit number 29–104).

Reviewer #1:

Comments 1-3

• Abstract: "flexibly" should be "properly"

• Abstract: "remained impaired" should be "remained different" You have no evidence of impairment, just differences in axis orientation.

• Line 149: "cares" should be "care"

Answer to comments 1-3

Thank you for kindly pointing them out. Following to the reviewer #1’s comments, we have revised the manuscript.

General comment

The remodeling of fracture callus (woven bone) into lamellar bone is affected by loading since osteonal orientation is affected by bone stress (see, for example, "Skeletal Tissue Mechanics Edition 2" Martin et al, discussion around Figure 6.14). A discussion of this is not necessary, but would make the results of the paper fit better into existing literature. My conclusion from your experiment is that stress shielding under a plate that is insufficient to cause an effect on BMD, can still affect the details of consolidation of the woven bone of the fracture callus (i.e., collagen and mineral long-axis directions) by affecting paths of osteonal cutting cones and, perhaps, how the osteoblasts deposit matrix.

Answer to general comment

Thank you for your valuable comment. As the reviewer #1 suggested, the orientation of osteon in which collagen preferentially orients along its longitudinal axis [Ascenzi A, Bonucci E. The tensile properties of single osteons. Anat Rec. 1967; 158: 375-386] should be important for the formation of preferential collagen/apatite orientation. With the reviewer #1’s comment, we felt that discussing the relationship between stress shielding and apatite orientation, by citing the existing literatures on the directional relationship between osteon and bone stress, reinforces our claim. Therefore, we have added the following sentences and references in Discussion section. 

P. 12, line 259

This unrecovered apatite orientation under stress shielding might be influenced by the osteon orientation. The orientation of osteon in which collagen preferentially orients along its longitudinal axis [29] corresponds with the directions of the maximum principal stress [30, 31]. Under the stress shielding, the direction of osteon would be disturbed, which further led to the low degree of collagen orientation. Since the apatite crystallizes on collagen so that its c-axis aligns with the long axis of collagen in the presence of osteocalcin [17], the apatite c-axis orientation degrades by inheriting degraded collagen orientation.

Added references:

29. Heřt J, Fiala P, Petrtýl M. Osteon orientation of the diaphysis of the long bones in man. Bone. 1994; 15: 269-277.

30. Martin RB, Burr DB, Sharkey NA, Fyhrie DP. Skeletal tissue mechanics. 2nd ed. New York: Springer; 2015.

31. Ascenzi A, Bonucci E. The tensile properties of single osteons. Anat Rec. 1967; 158: 375-386.

Reviewer #2:

The authors presented a high-quality study of the effect of a single fracture plate for use in veterinary applications. The authors demonstrated reduced BMD and, more so, reduced apatite c-axis preferential orientation in the vicinity of the fracture site and corresponding surrounding bone plate. This reviewer does not have any additional questions.

Answer to comment

Thank you for your review and your encouraging comment. We will continue to devote ourselves to research activities for the development of veterinary medicine and science.

We hope that the revised manuscript is suitable for publication in PLOS ONE.

Sincerely yours,

Hideo Akiyoshi

---

## [Editor Report · Decision Letter 1]

31 Jul 2020

Impaired bone quality characterized by apatite orientation under stress shielding following fixing of a fracture of the radius with a 3D printed Ti-6Al-4V custom-made bone plate in dogs

PONE-D-20-18343R1

Dear Dr. Akiyoshi,

We’re pleased to inform you that your manuscript has been judged scientifically suitable for publication and will be formally accepted for publication once it meets all outstanding technical requirements.

Kind regards,

Ryan K. Roeder, PhD

Academic Editor

PLOS ONE
---

## [Editor Report · Acceptance letter]

24 Aug 2020

PONE-D-20-18343R1 

Impaired bone quality characterized by apatite orientation under stress shielding following fixing of a fracture of the radius with a 3D printed Ti-6Al-4V custom-made bone plate in dogs 

Dear Dr. Akiyoshi:

I'm pleased to inform you that your manuscript has been deemed suitable for publication in PLOS ONE. Congratulations! Your manuscript is now with our production department. 

Kind regards, 

on behalf of

Dr. Ryan K. Roeder 

Academic Editor

PLOS ONE